# Insecticidal Activity of *Bacillus thuringiensis* Strains on the Nettle Caterpillar, *Euprosterna elaeasa* (Lepidoptera: Limacodidae)

**DOI:** 10.3390/insects11050310

**Published:** 2020-05-15

**Authors:** Angelica Plata-Rueda, Hughes Antonio Quintero, José Eduardo Serrão, Luis Carlos Martínez

**Affiliations:** 1Department de Entomology, Federal University of Viçosa, Viçosa, Minas Gerais 36570–000, Brazil; angelicaplata@yahoo.com.mx; 2Department of Crop Protection, Monterrey Oil Palm Plantation, Puerto Wilches, Santander 687–061, Colombia; quinterohughes408@gmail.com; 3Department of General Biology, Federal University of Viçosa, Viçosa, Minas Gerais 36570–000, Brazil; jeserrao@ufv.br

**Keywords:** anti-feeding effect, biopesticide, biological control, oil palm pest, survivorship, toxicity

## Abstract

In the present work, we evaluated the insecticidal activity of *Bacillus thuringiensis* (*Bt*) strains on *Euprosterna elaeasa* as an alternative for the organophosphate insecticide use in oil palm plantations in the Americas. The toxic effects of four *Bt*-strains (HD-1 var. *kurstaki*, SA-12 var. *kurstaki*, ABTS-1857 var. *aizawai*, and GC-91 var. *aizawai*) were evaluated against *E. elaeasa* caterpillars for toxicity, survival, anti-feeding, and mortality in field-controlled conditions. The *Bt*-strains, ABTS-1857 var. *aizawai* (LC_50_ = 0.84 mg mL^−1^), GC-91 var. *aizawai* (LC_50_ = 1.13 mg mL^−1^), and HD-1 var. *kurstaki* (LC_50_ = 1.25 mg mL^−1^), were the most toxic to *E. elaeasa*. The caterpillar survival was 99% without exposure to *Bt*-strains, and decreased to 52–23% in insects treated with the LC_50_ and 10–1% in insects exposed to LC_90_ after 48 h. Furthermore, *Bt*-strains decreased significantly the consumption of oil palm leaves of *E. elaeasa* 3 h after exposure. Mortality of *E. elaeasa* caterpillars caused by *Bt*-strains had similar lethal effects in the laboratory and in field conditions. Our data suggest that *Bt*-strains have insecticidal activity against *E. elaeasa* and, therefore, have potential applications in oil palm pest management schemes.

## 1. Introduction

The nettle caterpillar, *Euprosterna elaeasa* Dyar (Lepidoptera: Limacodidae) is a significant pest of *Elaeis guineensis* Jacquin (Arecales: Arecaceae) from Brazil, Colombia, Ecuador, Guyana, Mexico, Panamá, Peru, Surinam, Trinidad and Tobago, and Venezuela [1,2]. This insect also damages other palm trees species, such as *Bactris gasipaes* Kunth, *Calappa botryophora* (Mart.) Kuntze, *Cocos nucifera* Linnaeus, and *Desmoncus polyacanthos* (Mart.) Kuntze [1,3]. The life cycle of *E. elaeasa* is 64 days (egg 5.1, larva 35.2, pupa 19.4, and adult 4.7) [4]. *Euprosterna elaeasa* damages oil palm leaves with a consumption rate of 66 cm^2^/caterpillar, and the damage causes an 80% loss of plant canopy with 1000 insects/leaf. It is also a reason behind Pestalotiopsis fungal disease in oil palm plantations [2,4].

In Colombia, chemical insecticides such as acephate, methamidophos, and monocrotophos are used on oil palm crops to control *E. elaeasa* [1,5]. Due to the high level of infestation and the rapid spread of *E. elaeasa* in oil palm trees, the use of insecticides is common practice [6,7]. However, recent studies have shown the presence of these insecticides in minimal quantities in palm oil [8,9]. Conventional insecticides are expensive and cause environmental pollution [10], atmosphere ozone-depletion [11], residual long [12], and insecticide resistance [13]. New alternatives that are more sustainable, different from organophosphates, are needed to replace the main insecticides historically used against *E. elaeasa* for the past 50 years [1,6]. The search for alternatives for *E. elaeasa* control is important, considering the impact generated by the use of insecticides in this agroecosystem [6]. Thus, the use of natural enemies, such as viruses, bacteria, and fungi, can be an alternative for oil palm pest control [14,15,16].

*Bacillus thuringiensis* (*Bt*) is a biocontrol agent for defoliating pests worldwide, and individual strains are specific to a small group of insect targets without effects on animals and environment [17]. *Bt* is a gram-positive spore-forming bacterium with entomopathogenic properties. In the sporulation, *Bt* produces crystalline or “Cry” inclusions, called δ-endotoxins, biosynthesized during the second phase of the growth cycle [18]. In this cycle, the Cry proteins are converted in active toxins upon insect ingestion [19]. Several Cry proteins displaying activity on insects have been identified: the Cry1 proteins are toxic to Lepidoptera [20], while the Cry3 proteins are toxic to Coleoptera [21,22]; also, a high number of different subgroups (Cry1Ac, Cry1Ba, Cry8Ca, Cry1Eb, Cry1J, etc.) are active against mosquitoes, Coleoptera, Diptera, Hemiptera, and Hymenoptera [21,23,24].

*Bt* was reported as a biological control agent for oil palm pests [6,25]. Different oil palm lepidopteran species may have different levels of susceptibility to a specific Cry protein that occurs in *Metisa plana* Walker (Psychidae) [16], *Opsiphanes cassina* Felder (Nymphalidae) [26], and *Tirathaba rufivena* Walker (Pyralidae) [27]. This microbiological agent provides biodiversity in agroecosystems and the delivery of ecosystem services to agricultural production, especially in pest population regulation [28]. Since the entomopathogenic bacterium infects their host through the midgut, they hold greater potential as biocontrol agents for *E. elaeasa*; however, the use of *Bt* on this insect has not been carried out.

This study evaluated the insecticidal activity of *Bt* strains as potential agents to control *E. elaeasa*, explained in different experiments: (i) toxicity test, (ii) survivorship, (iii) anti-feeding effect, and (iv) mortality in field conditions. The objective was to contribute to the development of strategies for controlling *E. elaeasa*, as a replacement for organophosphate insecticides.

## 2. Materials and Methods

### 2.1. Insects

In the field, 2527 adults of *E. elaeasa* (males = 1284, females = 1243) were captured manually during the day, in 5-yr-old commercial plantations of oil palm in the county of Puerto Wilches, Santander, Colombia (N 07°20’, W 73°54’). The insects were transferred in plastic trays (30 × 50 × 50 cm) with perforated lids for ventilation to the Entomology Laboratory of the Oil Palm Monterrey Plantation (Puerto Wilches, Santander, Colombia) to establish a breeding colony in laboratory conditions. Adults were fed a honey solution daily (15 mL of honey and distilled water, in a 2:1 ratio) applied with a sponge. Males and females of *E. elaeasa* were isolated in glass containers (30 × 30 × 30 cm) covered with a nylon mesh and containing *E. guineensis* leaves. For egg development, 9800 eggs oviposited on the surface of the leaves were collected every 24 h and placed in Petri dishes (90 × 15 mm high) containing a paper towel saturated with water. After hatching, first-instar caterpillars (*n* = 7550) were placed individually in glass vials (5 × 25 cm) covered with cotton and fed every 24 h with *E. guineensis* leaves. Eggs and caterpillars were maintained in incubators at 27 ± 1 °C, with 75 ± 5% RH and 12:12 (L:D) photoperiod. Newly third instar *E. elaeasa* caterpillars were used in the laboratory and field condition bioassays.

### 2.2. Concentration–Mortality Bioassay

Commercial *Bt* formulations commonly used to control Lepidoptera were used in all bioassays and selected for quality, high-efficiency, and non-toxicity (toxicity Class IV) [18]. The following *Bt* strains, HD-1 var. *kurstaki* (Dipel^®^, Abbott Laboratories, North Chicago, IL, USA), SA-12 var. *kurstaki* (Thuricide^®^, Certis USA LLC, Columbia, MD, USA), ABTS-1857 var. *aizawai* (XenTari^®^, Valent Bioscience Corporation, Osage, Iowa, USA), and GC-91 var. *aizawai* (Agree^®^, Certis USA LLC, Columbia, MD, USA), were prepared in an aqueous solution with 0.1% Triton X-100 (strains and distilled water) to obtain a stock suspension (100 g L^−1^), from which dilutions were prepared as needed. Six concentrations (0.156, 0312, 0.625, 1.25, 2.5, and 5 mg mL^−1^) were used to evaluate the toxicity of each *Bt*-strain to *E. elaeasa* caterpillars, construct concentration–mortality curves, and estimate the lethal concentrations (LC_50_ and LC_90_). Distilled water with 0.1% Triton X-100 was used as control. The application of the concentrations was carried out by the feeding method using oil palm leaves. Pieces (10 × 10 mm) of oil palm leaves were cut, sterilized with 5% sodium hypochlorite with three successive series of distilled water, and dried at room temperature. Then, pieces of oil palm leaf were dipped in solutions of different concentrations of each *Bt*-strain for 10 s and allowed to air dry for a period of 1 h. Caterpillars were placed individually in Petri dishes, and a piece of oil palm leaf treated with *Bt*-strain was provided. Three replicates with 50 insects of each were used in concentration testing, and the experimental design was completely randomized. The dead insects were counted after 48 h *Bt*-strain exposure.

### 2.3. Time–Mortality Bioassay

Caterpillars of *E. elaeasa* were placed individually in Petri dishes and exposed to the lethal concentrations (LC_50_ and LC_90_) of each of the *Bt*-strains determined in the dose–response relationship. A control was performed using distilled water with 0.1% Triton X-100. Exposure procedures and conditions were the same as described above for the concentration–mortality bioassay Section 2.2. The number of live insects was recorded every 6 h for 2 d. Three replicates of 50 insects were used by each *Bt*-strain and the experimental design was completely randomized.

### 2.4. Anti–Feeding Effect

Caterpillars of *E. elaeasa* were placed individually in Petri dishes with a piece of oil palm leaf (10 × 10 mm) treated with LC_50_ or LC_90_ of each *Bt*-strains and distilled water as control. Caterpillars were in contact with *E. guineensis* leaves for 3 h and, after this, the pieces were photographed with a digital photographic camera (D40, 18D55 mm, Nikon Corporation, Tokyo, Kantō, Japan) with a 15 cm macro focus in natural and flourishing light (SB-700 Nikon Corporation). The images were analyzed using the digital analysis software, UTHSCSA Image Tool v. 2.0 (University of Texas, Austin, TX, USA). The leaf area consumed by the caterpillar was measured in mm^2^, with pixels based on the RGB (red, 213 nm; green, 111 nm; blue, 56 bits) histogram. Twenty repetitions for each of the *Bt*-strain concentrations (LC_50_ and LC_90_) and control were carried out in a completely randomized design.

### 2.5. Mortality in Semi–Controlled Test

The bioassay was conducted in 5-yr-old commercial oil palm plantations (cv ‘Tenera’ × ‘Deli Ghana’) in the county of Puerto Wilches (Santander, Colombia), with an average temperature of 27.98 °C, 81–93% relative humidity, 1455 to 2258 h of sunshine per year, and 2189 mm of annual rainfall. In these conditions, fifty palm trees were selected and *E. elaeasa* caterpillars were used for each *Bt*-strain in the controlled field test. For each palm tree, 50 caterpillars were placed on the leaf No. 17, according to the rules of phyllotaxy [29] and isolated with a nylon trap (0.5 × 0.5 × 1.20 m). Treatments consisted of adding each *Bt*-strain at the calculated LC_90_ concentration, and distilled water with 0.1% Triton X-100 as the control, with ten replications per treatment. Applications of 100 mL of *Bt*-strain per leaf were made by a manual pump (Royal Condor^®^, 5 L capacity, Soacha, Cundinamarca, Colombia), and the number of dead caterpillars was counted after 15 d *Bt*-strain exposure.

### 2.6. Statistical Analysis

The concentration–mortality data were submitted to Probit analysis to obtain a dose–response curve [30]. The time–mortality data were analyzed for survival analysis (Kaplan-Meier estimators, log-rank test) with the Origin Pro 9.1 software (OriginLab Corporation, Northampton, MA, USA). Anti-feeding effect data were arcsine-transformed and submitted to one-way ANOVA, and a Tukey’s honestly significance difference (HSD) (*p* < 0.05) test was also used for comparison of means. Mortality data in semi-controlled conditions were summarized in percentages and submitted to one-way ANOVA and a Tukey’s HSD (*p* < 0.05); also, all values presented as mean ± SEM. Statistical procedures were analyzed by SAS 9.0 software (SAS Institute, Campus Drive Cary, NC, USA).

## 3. Results

### 3.1. Concentration–Mortality Bioassay

The dose–response model provided a good fit to the data (*p* > 0.05), allowing the determination of toxicological endpoints, and confirms the toxicity of each *Bt*-strain to *E. elaeasa*
Table 1. The bioassay showed that ABTS-1857 var. *aizawai* had LC_50_ = 0.84 mg mL^−1^ (range of 0.66–1.16 mg mL^−1^), GC-91 var. *aizawai* had LC_50_ = 1.09 mg mL^−1^ (range of 0.74–1.72 mg mL^−1^), HD-1 var. *kurstaki* had LC_50_ = 1.13 mg mL^−1^ (range of 0.84–1.56 mg mL^−1^), and SA-12 var. *kurstaki* had LC_50_ = 1.25 mg mL^−1^ (range of 0.80–2.13 mg mL^−1^). Mortality was <1% in the control group.

### 3.2. Time–Mortality Bioassay

Survival rate was determined 48 h after *E. elaeasa* caterpillar exposure to *Bt*-strains at lethal concentrations, LC_50_ and LC_90_. Survival rates differed between treatments at LC_50_ (log-rank test, χ^2^ = 9.47, df = 3, and *p* < 0.001). *E. elaeasa* survival decreased from 99.9% in the control to 52.79% with SA-12 var. *kurstaki*, 51.37% with GC-91 var. *aizawai*, 35.62% with HD-1 var. *kurstaki*, and 23.12% with ABTS-1857 var. *aizawai* (Figure 1A).

At LC_90_, the survival rates of *E. elaeasa* were different according to the treatments (log-rank test, χ^2^ = 18.57, df = 4, *p* < 0.001), decreasing from 99.9% (control) to 10.13% with SA-12 var. *kurstaki*, 9.87% with GC-91 var. *aizawai*, and 0% with both the HD-1 var. *kurstaki* and ABTS-1857 var. *aizawai* (Figure 1B).

### 3.3. Anti–Feeding Effect

The four *Bt*-strains caused an anti-feeding effect on *E. elaeasa* caterpillars, with lower leaf area consumed in comparison to control (Figure 2). The leaf area consumed by *E. elaeasa* differed between *Bt*-strains at LC_50_ (F_4,19_ = 9.51, *p* < 0.001), decreasing from 26.41 mm^2^ (control) to 11.38 mm^2^ with GC-91 var. *aizawai*, 8.89 mm^2^ with ABTS-1857 var. *aizawai*, 8.44 mm^2^ with SA-12 var. *kurstaki*, and 6.83 mm^2^ with HD-1 var. *kurstaki* (Figure 2A). However, at LC_90_ all *Bt*-strains had similar anti-feeding effects among them (F_4,19_ = 27.36, *p* > 0.05; Figure 2B).

### 3.4. Mortality in Semi–Controlled Test

The mortality caused by the *Bt*-strains to *E. elaeasa* caterpillars was different in a semi–controlled test (F_4,49_ = 48.19; *p* < 0.05), as shown in Figure 3. Mortality caused by *Bt*-strains of LC_90_ to *E. elaeasa* caterpillars was higher in HD-1 var. *kurstaki* and ABTS-1857 var. *aizawai* (92.1 ± 0.2% and 89.1 ± 2.1%, respectively) than with GC-91 var. *aizawai* and SA-12 var. *kurstaki* (86.8 ± 5.1% and 84.1 ± 4.9%, respectively), but they were all higher than in the control (2.67 ± 0.7%).

## 4. Discussion

The use of various *Bt*-strains was effective in causing mortality, compromising survivorship, and reducing the consumption rate of the nettle caterpillar, *E. elaeasa*. The *Bt*-strains HD-1 var. *kurstaki*, SA-12 var. *kurstaki*, ABTS-1857 var. *aizawai*, and GC-91 var. *aizawai* were toxic to *E. elaeasa* caterpillars and have a strong effect through oral exposure. *Bt*-strains caused mortality in *E. elaeasa* in a concentration-dependent manner, as demonstrated in other defoliating pests [19,31,32]. *Euprosterna elaeasa* caterpillars exposed to high concentrations of *Bt*-strains displayed muscle contractions, oral or anal secretions, and consequently, septicemia. In this context, symptoms in *E. elaeasa* caterpillars were consistent with the known effects of microbial disruption of insect midgut membranes. A set of results point to the effects on the digestive system of lepidopterous pests, such as *Diatraea saccharalis* Fabricius (Crambidae) [33], *Plutella xylostella* Linnaeus (Plutellidae) [34], and *Spodoptera frugiperda* JE Smith (Noctuidae) [35], after *Bt*-strains oral exposure. In general, few *Bt*-strains are effective against *E. elaeasa* at different concentrations and reinforce their use as an alternative to organophosphate insecticides on this species.

In this study, the survival time of *E. elaeasa* decreases mainly with HD-1 var. *kurstaki* and ABTS-1857 var. *aizawai*. However, periods of *Bt*-strains, from 24 to 48 h, were necessary to induce mortality in *E. elaeasa*. Survivorship of this insect is associated with the quick action in the midgut of several Cry proteins produced by *Bt*-strains and observed in other lepidopteran pests [36,37]. In this study, the compared effects of the *Bt*-strains on *E. elaeasa* occur at various periods. These time differences occur commonly between strains of the same subspecies (e.g., *Bt* var. *kurstaki* and *Bt* var. *aizawai*) [38], by specific variation of the δ-endotoxins originating from different *Bt*-strains [39], host immune responses [40], and virulent factors of strain types [41]. In this sense, *Bt* toxins have been reported to reduce or inhibit larval growth, development, or weight, interrupting the insect’s lifecycle [42]. The rapid effect against *E. elaeasa* suggests that the insecticidal activity of *Bt*-strains causes detrimental effects on neonates, with appreciable population reduction during the first days of infestation and can be essential for protecting oil palm trees.

The decrease in the consumption of oil palm leaves treated with LC_50_ and LC_90_ of *Bt*-strains suggests an anti-feeding effect on *E. elaeasa*, represented by different rates of intoxication and, consequently, cessation of feeding. On the other hand, the concentration-dependent effect on total food consumption by both lethal concentrations indicates that intoxication of *Bt*-strains is cumulative. Effects of *Bt*-strains after 24 h exposure on *Helicoverpa armigera* Hübner (Noctuidae), *Phyllocnistis citrella* Stainton (Gracillariidae), and *Tuta absoluta* Meyrick (Gelechiidae) were observed [43,44,45], causing a dramatic reduction in initial leaf consumption. In *E. elaeasa*, the reduced consumption (in both lethal concentrations) exposure regimes demonstrated that intoxication has serious deleterious effects that may translate into metabolic costs associated with the repair of midgut epithelium damage in caterpillar survivors. For instance, altered permeability and damage of the midgut interfere with food uptake, affect activity enzymes associated with digestion, and influence energy metabolism. Additionally, intoxication alters hemolymph pH and suppresses the immune response [19,39,40]. Our results demonstrate that the interplay of concentration and exposure regimen can produce anti-feeding effects, indicating that *Bt*-strains intoxication of *E. elaeasa* caterpillars is cumulative.

The *Bt*-strains, HD-1 var. *kurstaki*, SA-12 var. *kurstaki*, ABTS-1857 var. *aizawai*, and GC-91 var. *aizawai*, showed lethal effects against *E. elaeasa* in palm trees in the field, and results were consistent with those observed in the laboratory. However, mortality level at the larval stage was lower than those obtained under laboratory conditions. Efficacy of *Bt*-strains in field conditions may be due to environmental factors [46], toxins degradation [47], gut microbiota competition [48], and inactivation by the target organism [49]. The lethal effect of *Bt* and its effectiveness was also studied in other Limacodidae pests under field conditions as a potential biocontrol agent for *Acharia apicalis* Dyar [50], *Acharia fusca* Stoll [51], and *Parasa lepida* Cramer [52]. The results show that *Bt*-strains have a specific mode of action that affects a high number of *E. elaeasa* caterpillars. In particular, HD-1 var. *kurstaki* and ABTS-1857 var. *aizawai* are the most effective in the field, and that maximum efficiency from strains should be used during this life stage.

## 5. Conclusions

The insecticidal effect of four *Bt*-strains for controlling *E. elaeasa* were studied. The *Bt*-strains HD-1 var. *kurstaki*, SA-12 var. *kurstaki*, ABTS-1857 var. *aizawai*, and GC-91 var. *aizawai* cause mortality, reduces survivorship, and an anti-feeding effect on this insect, with the potential to control its field populations. The toxicity of the bacterium may efficiently manage *E. elaeasa* caterpillars and reduce the insect’s damage to oil palm trees and transmission of the Pestalotiopsis fungal complex. *Bt*-strains have lethal and sublethal effects on *E. elaeasa,* and are an alternative to organophosphate insecticides in oil palm plantations, aiding efforts to manage insecticide resistance.

## Figures and Tables

**Figure 1 insects-11-00310-f001:**
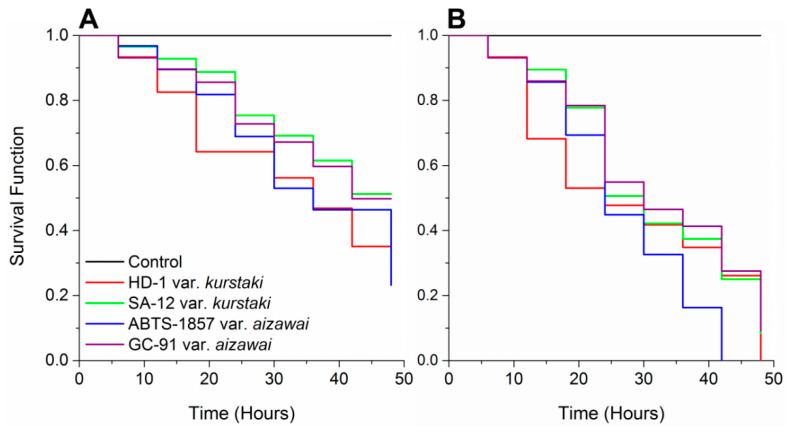
Survival curves of *Euprosterna elaeasa* caterpillars exposed to *Bacillus thuringiensis* strains, subjected to survival analyses using the Kaplan–Meier estimators’ log-rank test. Lethal dose of (**A**) LC_50_ (χ^2^ = 9.47; *p* < 0.001) and (**B**) LC_90_ (χ^2^ = 18.57; *p* < 0.001).

**Figure 2 insects-11-00310-f002:**
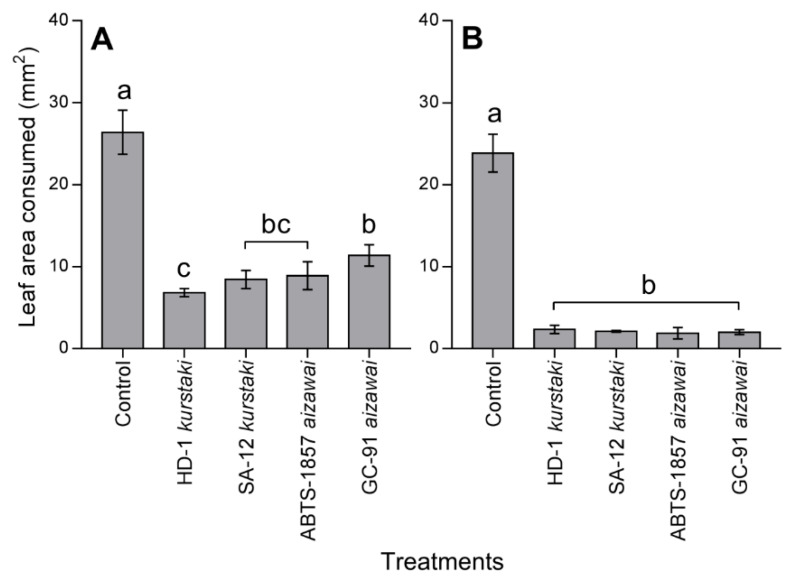
Leaf area consumed by *Euprosterna elaeasa* caterpillars exposed to *Bacillus thuringiensis* strains (LC_50_ (**A**) and LC_90_ (**B**) estimated values). Treatments (mean ± SEM) differ at *p* < 0.05 (Tukey’s mean separation test).

**Figure 3 insects-11-00310-f003:**
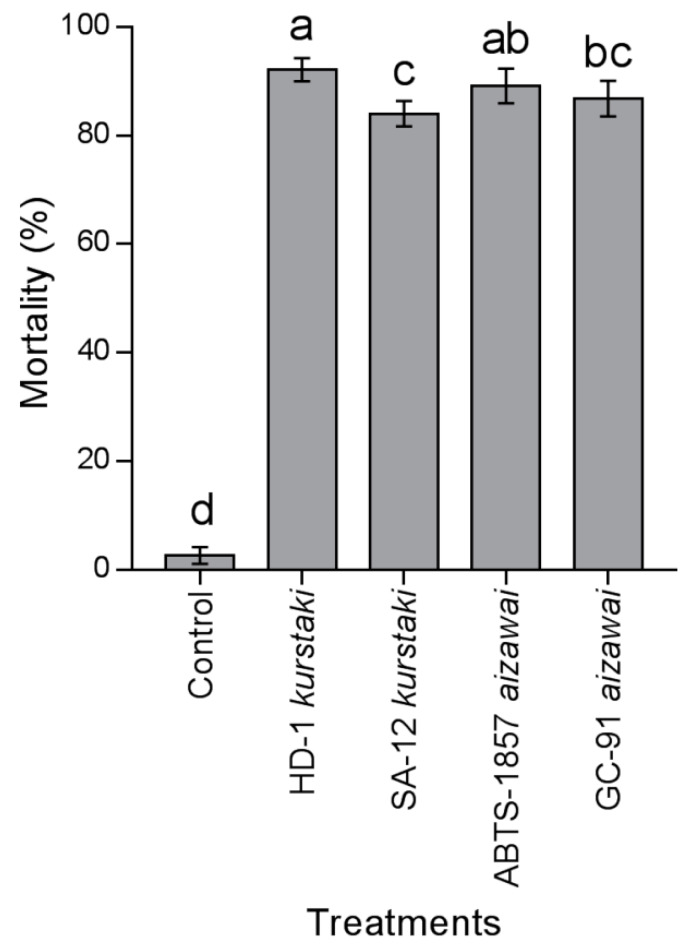
Mortality of *Euprosterna elaeasa* caterpillars by *Bacillus thuringiensis* strains to level LC_90_ application on oil palm trees. Treatment means (percent mortality ± SEM) with different letters show significant differences by Tukey’s HSD test at the *p <* 0.05 level.

**Table 1 insects-11-00310-t001:** Lethal concentration of *Bacillus thuringiensis* strains against *Euprosterna elaeasa* after 48 h exposure obtained from probit analysis (df = 5). The chi-square value refers to the goodness of fit test at *p* > 0.05.

Strain	No. Insects	Lethal Concentration	Estimated Concentration (mg mL^−1^)	95% Confidence Interval (mg mL^−1^)	Slope ± SE	χ^2^ (*p*-Value)
HD-1 var. *kurstaki*	150	LC_50_	1.133	0.845–1.561	2.22 ± 0.25	1.23 (0.36)
	150	LC_90_	4.268	2.802–8.512		
SA-12 var. *kurstaki*	150	LC_50_	1.258	0.805–2.136	2.40 ± 0.41	1.89 (0.16)
	150	LC_90_	4.299	2.442–10.92		
ABTS-1857 var. *aizawai*	150	LC_50_	0.840	0.664–1.075	1.73 ± 0.35	1.34 (0.22)
	150	LC_90_	4.623	3.172–7.875		
GC-91 var. *aizawai*	150	LC_50_	1.097	0.742–1.724	2.40 ± 0.41	1.38 (0.22)
	150	LC_90_	4.579	2.647–8.894		

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
