# Peer review of "Insecticidal Activity of *Bacillus thuringiensis* Strains on the Nettle Caterpillar, *Euprosterna elaeasa* (Lepidoptera: Limacodidae)"

_insects, 2020, doi:10.3390/insects11050310_

Round 1
Reviewer 1 Report
The authors have obtained some results with Bt strains that should be of use to the palm oil industry. The data is well presented, and the English is good for the Introduction and Methods, but breaks down some in the Results section, always the hardest to write and edit.
I have a few editorial suggestions:
- The authors do not really explain why they chose the four strains of Bt that they tested. This should be done in the Introduction or beginning of the Results section. This is especially needed for the reason in the next item.
- The authors say in the Discussion section (paragraph 2) that their results are highly variable with the Bt strain, but to me they look quite similar. The LC50s and especially the LC90s are almost identical. Why? Do these strains have the same Cry proteins?
- in the methods subsection, 2.4 Antifeeding, the authors indicate they are using soybean leaves. I assume this is an oversight.
- I suggest the authors use Grammar check in MS Word to proof their paper. There are a number of bad sentences in the Results section and a few elsewhere.
Author Response
Please find enclosed our revised manuscript entitled “Insecticidal Activity of Bacillus thuringiensis Strains on the Nettle Caterpillar, Euprosterna elaeasa (Lepidoptera: Limacodidae)” (No. 784796_R1). The comments and suggestion provided were greatly appreciated and carefully considered for preparing the current version. The main changes are red-marked in the manuscript, and the point-by-point explanations to the comments provided follow below:
Reviewer #1
The authors have obtained some results with Bt strains that should be of use to the palm oil industry. The data is well presented, and the English is good for the Introduction and Methods, but breaks down some in the Results section, always the hardest to write and edit.
I have a few editorial suggestions:
The authors do not really explain why they chose the four strains of Bt that they tested. This should be done in the Introduction or beginning of the Results section. This is especially needed for the reason in the next item.
The authors say in the Discussion section (paragraph 2) that their results are highly variable with the Bt strain, but to me they look quite similar. The LC50s and especially the LC90s are almost identical. Why? Do these strains have the same Cry proteins?
in the methods subsection, 2.4 Antifeeding, the authors indicate they are using soybean leaves. I assume this is an oversight.
I suggest the authors use Grammar check in MS Word to proof their paper. There are a number of bad sentences in the Results section and a few elsewhere.

Reviewer 2 Report
The manuscript presents the results of experiments conducted in the lab and in semi-field conditions with different Bt strains against Euprosterna elaeasa.
The manuscript is generally well organized and the results are significant.
There are several words and phrases that need to be fixed for English language.
Some specific comments are below:
- line 50: change "have" to "has"
- lines 209-210: this sentence is not clear; explain better. Then, this study does not investigate directly the effects on the gut histopathology to drown such conclusions. This should be appropriately weighed.
- line 223: change "food" to feeding"
Lines 248 - on: while the laboratory results are very significant and the semi-field trials are encouraging, this study does not provide data on the efficacy in larger scale field conditions, so that any conclusion on the use in biological control should be revised and appropriately weighed in the present manuscript.
Author Response
Please find enclosed our revised manuscript entitled “Insecticidal Activity of Bacillus thuringiensis Strains on the Nettle Caterpillar, Euprosterna elaeasa (Lepidoptera: Limacodidae)” (No. 784796_R1). The comments and suggestion provided were greatly appreciated and carefully considered for preparing the current version. The main changes are red-marked in the manuscript, and the point-by-point explanations to the comments provided follow below:
Reviewer #2:
The manuscript presents the results of experiments conducted in the lab and in semi-field conditions with different Bt strains against Euprosterna elaeasa.
The manuscript is generally well organized and the results are significant.
There are several words and phrases that need to be fixed for English language.
Some specific comments are below:
- line 50: change "have" to "has"
- lines 209-210: this sentence is not clear; explain better. Then, this study does not investigate directly the effects on the gut histopathology to drown such conclusions. This should be appropriately weighed.
- line 223: change "food" to feeding"
Lines 248 - on: while the laboratory results are very significant and the semi-field trials are encouraging, this study does not provide data on the efficacy in larger scale field conditions, so that any conclusion on the use in biological control should be revised and appropriately weighed in the present manuscript.
